# Physiological, Psychological and Performance-Related Changes Following Physique Competition: A Case-Series

**DOI:** 10.3390/jfmk5020027

**Published:** 2020-04-25

**Authors:** Jaymes M. Longstrom, Lauren M. Colenso-Semple, Brian J. Waddell, Gianna Mastrofini, Eric T. Trexler, Bill I. Campbell

**Affiliations:** Exercise Science Program, University of South Florida, Performance and Physique Enhancement Laboratory, Tampa, FL 33620, USA; jlongstrom@mail.usf.edu (J.M.L.); lcolensosemple@gmail.com (L.M.C.-S.); bwaddell@mail.usf.edu (B.J.W.); gfmastrofini@mail.usf.edu (G.M.); ericttrexler@gmail.com (E.T.T.)

**Keywords:** body composition, metabolic rate, bodybuilding, weight gain, physique

## Abstract

The purpose of this case-series was to evaluate the physiological, psychological and performance-related changes that occur during the postcompetition period. Participants included three male (34.3 ± 6.8 years, 181.6 ± 8.9 cm) and four female (29.3 ± 4.9 years, 161.4 ± 6.0 cm) natural physique athletes. Body composition (fat mass (FM) and fat-free mass (FFM); Skinfold), resting metabolic rate (RMR; indirect calorimetry), total body water (TBW; bioelectrical impedance analysis), sleep quality (PSQI; Pittsburgh Sleep Quality Index), quality of life measures (RAND SF36), menstrual irregularities, and knee extension performance were assessed 1–2 weeks prior to competition, and 4 weeks and 8–10 weeks postcompetition. Blood hormones (free triiodothyronine; T_3_, free thyroxine; T_4_, and leptin) were assessed at 1–2 weeks prior to competition and 8–10 weeks postcompetition. Participants tracked daily macronutrient intake daily for the duration of the study. Group-level data were analyzed using exploratory, one-tailed, nonparametric statistical tests. Bodyweight, FM, bodyfat%, RMR, and blood hormones (T_3_, T_4_, and leptin) increased significantly (*p* < 0.05) at the group level. Relative (%Δ) increases in fat mass were associated with △RMR (τ = 0.90; *p* = 0.001) and △leptin (τ = 0.68; *p* = 0.02), and △leptin was associated with △RMR (τ = 0.59; *p* = 0.03). The time course for recovery appears to vary substantially between individuals potentially due to strategies implemented postcompetition.

## 1. Introduction

During contest preparation, physique competitors attempt to reduce body fat while maintaining fat-free mass (FFM) by implementing a hypocaloric diet, participating in resistance training, and increasing aerobic exercise during a period known as contest preparation [1]. Contest preparation, which can last upwards of 20 weeks [2], is associated with hormonal dysregulation [2,3,4,5], loss of FFM [2,6,7], suppressed resting metabolic rate (RMR) [2,3,6,8] and downregulated non-resting energy expenditure [9,10]. These unfavorable physiological adaptations can last for several months postcompetition [2,3,8,11,12], contributing to episodes of overeating [11,13] that may result in accumulation of body fat above baseline levels, referred to as bodyfat overshooting. More importantly, physiological impairments are associated with an insufficient energy intake to meet the demands of energy expenditure, known as low-energy availability (LEA) [14]. Previous research demonstrates physique competitors routinely meet the criterion for LEA (<30 kcal/kg/FFM) during contest preparation [2,6,7,12]. This threshold of LEA has been shown to disrupt hormones like leptin and triiodothyronine (T_3_) in healthy women [15], along with a wide range of other physiological, psychological and performance-related consequences, which describes the criteria for a syndrome known as Relative Energy Deficiency in Sport (RED-S) [16].

Contest preparation is also associated with poor sleep, mood disturbances and exercise performance decrements [2,3,8]. Changes in sleep, mood disturbances and decreased dietary intakes during contest preparation could contribute to decreases in exercise performance [17]. Although exercise performance is not part of judging criteria in physique sport, resistance exercise is the primary modality for physique athletes to maintain or possibly improve muscle mass during contest preparation. If decrements in exercise performance are sustained it could be indicative of overtraining syndrome [18]. While some studies report favorable or stable mood scores leading up to competition [5,6], other studies demonstrate mood disturbances near competition that may persist for several months postcompetition [2,19].

The time course for recovery of physiological parameters is not well understood and appears to be highly variable between individuals [11]. RMR can return to baseline values in as little as five weeks [8,11] or upwards of five months [3] following competition. While some hormone levels are restored within 12 weeks, others such as T_3_, T_4_ and leptin can remain downregulated for several months [2,3,5,11]. Female competitors routinely experience menstrual cycle irregularities or amenorrhea during contest preparation [19,20], which can persist for up to a year postcompetition despite body fat regain and pre-diet energy availability [12]. 

While several case studies have evaluated drug-free competitors during the contest preparation period [2,3,6,7,8,12,19,20,21] and one case-series included a short postcompetition transition period [22], few have comprehensively examined the recovery phase from physique competition [2,3,8,12]. These investigations highlight the need for the development of effective strategies to help athletes reverse the unfavorable adaptations associated with contest preparation. However, single case study designs lack sufficient sample sizes to observe the heterogeneity in responses that occur during recovery from physique competition, therefore limiting general recommendations. Thus, the purpose of this study was to evaluate the changes to physiological, psychological and exercise performance parameters in three male and four female physique athletes during the postcompetition transition to the offseason. 

## 2. Materials and Methods 

### 2.1. Competitor Assessment

This case-series followed three male (M1, M2, M3; 34.3 ± 6.8 years, 181.6 ± 8.9 cm) and four female (F1, F2, F3, F4; 29.3 ± 4.9 years, 161.4 ± 6.0 cm) natural physique athletes during the postcompetition period following their final competition of the season. Participant characteristics are outlined in Table 1.

Data were collected at three time points: 1–2 weeks precompetition, four weeks postcompetition and 8–10 weeks postcompetition (Figure 1). Outcome measures included body composition, body water, metabolic rate, blood hormones, sleep quality, quality of life measures, menstrual irregularities and knee extension performance. All participants were assessed in the morning following an overnight fast, prior to ingestion of food or beverages, after abstaining from exercise for the prior 24 h. Prior to all assessments, participants were instructed to use the restroom to void their bladders. Self-reported dietary intakes were collected for the full postcompetition period. 

### 2.2. Body Weight and Body Water Assessment

Body height was measured on a physician beam scale (Health-O-Meter; model 402KL; Pelstar, Inc., McCook, IL, USA). Body weight, total body water, segmental impedance, extracellular and intracellular water were assessed via multifrequency bioelectrical impedance analysis (InBody® 570 Body Composition Analyzer; Biospace, Inc. Seoul, Korea). 

### 2.3. Assessment of Body Composition

Body composition was assessed via skinfold technique (Lange® caliper, Cambridge Scientific Industries, Inc, Cambridge, MD). Subcutaneous adipose thickness was measured at seven sites on the right side of the body according to ACSM guidelines [23]. Measurement sites included the triceps, chest, midaxilla, subscapula, suprailiac and thigh. Measurement sites were located using standard anatomical landmarks. For each skinfold assessment, the research assistant firmly grasped a double fold of skin between thumb and index finger, the caliper was placed directly on the skin surface, 1 cm away from the thumb and finger, perpendicular to the skinfold measuring the mm thickness of this fold. Three measurements were taken at each site with the average of the three recorded. Additional measurements were taken if measurements were not within 1–2mm. Once the fat thicknesses were recorded for each of the seven sites, body density was estimated using the Jackson-Pollock generalized skinfold equation [24] and percent body fat was estimated using the Siri equation [23]. Body composition was also evaluated using a three-compartment (3C) model, separating the body into fat mass (FM), FFM and water. Dry FFM was also estimated by subtracting total body water measurements from FFM measurements. This is to account for fluctuations in FFM following increased caloric intake. Each competitor’s body composition assessments were completed by the same technician, who’s calculated body fat percentage (BF%) test–retest reliability was intraclass correlation = 0.99; SEM = 0.196%; minimal detectable change = 0.54% fat.

### 2.4. Assessment of Sleep, Quality of Life Measures and Menstrual Cycle

Sleep quality was assessed with the Pittsburgh Sleep Quality Index (PSQI) survey [25]. The questionnaire includes 19 individual items that generate seven component scores; the sum of the seven component scores yields one global score to compare sleep quality changes. Higher scores indicate poorer sleep quality. Energy/fatigue, social functioning and emotional well-being were assessed utilizing the RAND 36-Item Health Survey 1.0. The survey consists of 36 questions that includes generic, coherent and easily administered quality-of-life measures. Each item is scored on a range of 0–100, with higher scores representing a more favorable health state, as defined by the RAND 36-Item Health Survey 1.0. Scores were than averaged together to create the energy/fatigue, social functioning and emotional well-being scales. Menstrual cycle irregularities were assessed in female participants utilizing a questionnaire with seven yes/no and short answer questions. 

### 2.5. Assessment of Metabolic Rate

Resting metabolic rate (RMR) was assessed with indirect calorimetry (TrueOne 2400 Canopy System, ParvoMedics, Sandy, UT, USA). The metabolic measurement system was calibrated prior to each assessment. RMR was assessed for twenty minutes. The first five minutes were discarded, and the remaining 15 min was averaged [26] for the calculation of RMR. Measured RMR values were also compared with predicted RMR values as estimated by the Harrison–Benedict equation [27]. During the test, participants were instructed to breathe normally and remain still, silent, relaxed and awake. The calculated test–retest reliability for the metabolic measurement system was intraclass correlation = 0.998; SEM = 25.6 kcals; minimal detectable change = 71kcals. 

### 2.6. Assessment of Knee Extension Performance

Muscular performance was assessed with three sets of knee extensions at 60% of the participant’s bodyweight measured during the first visit. Each repetition was performed with controlled eccentric and concentric contractions and a one second pause at full knee extension. All sets were performed to momentary muscular failure, operationally defined as the inability to perform another concentric repetition with proper form. Participants rested for one minute between sets. The number of repetitions and total weight lifted were recorded for each set. 

### 2.7. Assessment of Blood Hormones

Blood samples were obtained by certified phlebotomists via a cannula that was inserted into an antecubital vein, and analyzed for serum leptin, free triiodothyronine (T_3_) and free thyroxine (T_4_) at Quest Diagnostics Laboratory located at 3450 E. Fletcher Avenue, Suite250 Tampa, FL 33620. 

### 2.8. Nutritional Approach

Participants self-selected a postcompetition dietary strategy or followed the guidance of a contest preparation coach. Participants were instructed to record daily calorie, fat, carbohydrate, protein and fiber intake in a Microsoft Excel spreadsheet for two weeks before testing, prior to any peak week protocol and for the duration of the 8–10 week postcompetition period. Descriptive statistics for diet composition were calculated using Microsoft Excel.

### 2.9. Statistical Analysis

Group-level data were analyzed via one-tailed Wilcoxon-Pratt Signed-Rank tests and Kendall’s Rank Correlation tests at the α = 0.05 significance level. Group-level statistical tests were exploratory in nature and should therefore be used for the purpose of informing hypotheses for future studies, rather than forming conclusions. Changes from T1 to T3 were calculated for percent change (%△) in body weight, percent change in fat mass (%△FM) and raw change in body fat percentage (△bodyfat %). 

## 3. Results

### 3.1. Group Level Observations

At the group level, significant (all *p* < 0.05) increases were observed for bodyweight, FM, bodyfat%, RMR and blood hormones (T_3_, T_4_ and leptin). Fat-free mass increased but did not reach the threshold for statistical significance (*p* = 0.055). Percent change (%△) in body weight was associated with △RMR (tau (τ) = 0.62; *p* = 0.03) and △leptin (τ = 0.59; *p* = 0.03). %△FM was associated with △RMR (τ = 0.90; *p* = 0.001) and △leptin (τ = 0.68; *p* = 0.02). △bodyfat % was associated with △leptin (τ = 0.88; *p* = 0.003). △leptin was associated with △RMR (τ = 0.59; *p* = 0.03). 

### 3.2. Individual Data

Observed changes for all outcomes varied substantially between competitors. While the majority of competitors increased caloric intake to a similar degree, F1 increased caloric intake by 97% (Figure 2). This large increase in energy intake coincided with the greatest increase in body weight (△22%), while all other competitors increased body weight by less than 10% (Figure 2). Interestingly, all competitors increased fat mass and leptin, except for M2 and M3, who experienced a decrease in leptin with an increase in BF% (Figure 3). All competitors experienced increases in BF% from T1 to T3 that exceeded the minimal detectable change. A reduction in RMR was observed for F2 (−90kcal/day), which was greater than the minimal detectable change and M2 experienced no change. All other competitors experienced increases in RMR from T1 to T3 that exceeded the minimal detectable change (Figure 4). Individual responses for total number of completed reps during the knee extension test are depicted in Figure 5. Individual raw values for physiological outcomes are listed in Table 2. Weekly averaged diet composition values are listed in Table A1. Psychometric scores were variable and changed minimally in most competitors (Table A2).

## 4. Discussion

The purpose of this case-series was to evaluate physiological, psychological and performance-related changes in natural male and female physique athletes postcompetition. Bodyweight, FM, RMR and blood hormones (T_3_, T_4_ and leptin) increased significantly (*p* < 0.05) at the group level. Increases in fat mass percentage were strongly associated with increased RMR (τ = 0.90; *p* = 0.001) and leptin concentrations (τ = 0.68; *p* = 0.02). Exploratory analyses revealed a correlation between changes in bodyfat percentage and leptin levels (Figure 3). We observed improvements in a number of physiological parameters, knee extension performance and measures of sleep and quality of life. Our data, along with those reported previously, indicate that some measures improve substantially within several weeks [8,11], whereas others remain downregulated for upwards of nine months [2,5,12].

Although RMR is reportedly restored within four to six weeks postcompetition in some individuals [11], it can remain suppressed for as long as five months in others [2]. RMR increased above predicted values for most competitors. F2 experienced an unusual decrease in RMR, which exceeded the minimal detectable change indicating a real change occurred (F2; △RMR = −6%). This reduction in RMR is potentially attributable to exercise induced amenorrhea [28], as the participant reported menstrual cycle irregularities and performed high volume resistance training throughout the postcompetition period. M2 experienced no change in RMR, which coincided with a minimal increase in body weight and hormones, despite a 49% increase in caloric intake (Figure 2). Lack of weight gain and resultant suppressed RMR and hormones could potentially be explained by an insufficient increase in caloric intake to support high levels of self-reported activity energy expenditure from resistance exercise and a labor-intensive profession. M3 experienced an increase in RMR (△RMR = +10%) from T1 to T3 exceeding the minimal detectable change that could be explained by the apparent increase in dry FFM (+3kg)(Table 2) [29]. However, RMR did not exceed the predicted value, which could be explained by inadequate increases in T_3_ due to its association with metabolic rate [30]. 

Minimal increases in BF% by M2 and M3 likely contributed to inappreciable changes in leptin (Figure 3), since it appears notable changes in leptin occur following greater increases in BF% for a sustained period of time [31]. It is well established that fat mass is associated with leptin concentrations [9,32,33] and other research suggests a time course for leptin recovery in physique competitors of at least three months postcompetition [2,11]. Low levels of leptin and delayed recovery can contribute to postdiet hyperphagia due to impaired satiety signaling resulting in excessive body fat accumulation [34]. The rapid increase in body weight by F1 aligns with similar reports of body fat overshooting [35,36] and corresponded to a substantial increase in leptin (+3.2ng/mL) (Figure 3), which is potentially indicative of improved neuroendocrine and immune function [37]. However, body fat overshooting by F1 was likely due to low leptin levels achieved during contest preparation (0.5 ng/mL) and could negatively impact future fat loss efforts during contest preparation. In a tightly controlled overfeeding study by Johannsen and colleagues, 35 young overweight adults consumed 40% above their baseline energy requirements and found increases in fat mass to be proportionate to increases in leptin concentrations and significant increases in T_3_ but not T_4_ [38]. This supports the changes in fat mass and hormones experienced by F1 and the minimal but variable changes in T_3_ and T_4_ we observed in the other competitors in our study (Table 2). The primary objective for physique competitors in the offseason is to maximize increases in FFM [39] and delayed recovery in physiological outcomes can impair improvements in FFM [14]. 

Only one previous case study reported improvements in sleep quality during the postcompetition period [3]. While this is generally consistent with our observations, M2 experienced a decline in sleep quality, potentially explained by irregular sleep patterns due to shift work (Table A2). Interestingly F1 experienced a decrease in psychometric scores for energy/fatigue which coincided with a decrease in leg extension performance from T2 to T3 (Figure 5). This aligns with results reported by Rossow and colleagues suggesting that alterations in mood states could explain performance decrements [2]. Previous case studies have reported performance measures to remain suppressed for several months following competition [2,3,8]. However, we observed all competitors improved leg extension performance from T1 to T3, likely due to differences in assessments (Figure 5). All female competitors experienced menstrual cycle irregularities during postcompetition, which could take upwards of 70 weeks to return to normal, even when body fat levels have been restored [12]. F1 was the only female to display a normalized menstrual cycle during the postcompetition period, likely due to large increases in all measures. 

Importantly, the time course for recovery appears to vary substantially between individuals potentially due to strategies implemented postcompetition. A previous case study describes a strategy popularized in natural bodybuilding known as reverse dieting, in which competitors attempt to minimize weight gain by gradually increasing caloric intake in small increments while continuing to perform aerobic exercise [3]. Three competitors (M2, F2, F4) increased caloric intake with minimal weight gain (△BW%: +2%, +5%, +5%) (Figure 2). RMR changes were minimal (△RMR%: 0%, −6%, +4%) and dry FFM decreased (−0.6kg, −1kg, −0.5kg) (Table 2). It appears that such minor increases in body weight (< 5%) were inadequate to elicit improvements in physiological outcomes. Competitors could potentially expedite recovery by increasing caloric intake and reducing aerobic exercise to facilitate a more substantial rate of weight gain in the weeks following competition. 

Delayed time course of recovery should encourage longer periods between competitive seasons to not only allow for adequate recovery but improvements in key physiological variables during the offseason. Other researchers advocate for competing at most every other year [40]. The small sample size and lack of baseline measurements prior to contest preparation in the current study precludes generalizing findings to all physique competitors and results should be interpreted with caution. Future investigations are required to determine an appropriate rate of weight gain postcompetition to facilitate adequate recovery.

## 5. Conclusions

To our knowledge this is the first case-series to examine natural physique athletes for 8**–**10 weeks following competition, providing a comprehensive evaluation of physiological, psychological and performance-related changes. Low levels of body fat achieved during contest preparation coincide with substantial energy restriction leading to adverse consequences that may persist if interventions targeted to facilitate recovery are not implemented. Physique competitors expecting to optimally recover from competition should include a level of energy intake that elicits a certain amount of body fat gain, while avoiding body fat overshooting. Additionally, physique athletes are encouraged to incorporate longer offseasons not only to allow for proper recovery of key variables but to support improvements in lean body mass and metabolic rate for future competitive success. 

## Figures and Tables

**Figure 1 jfmk-05-00027-f001:**
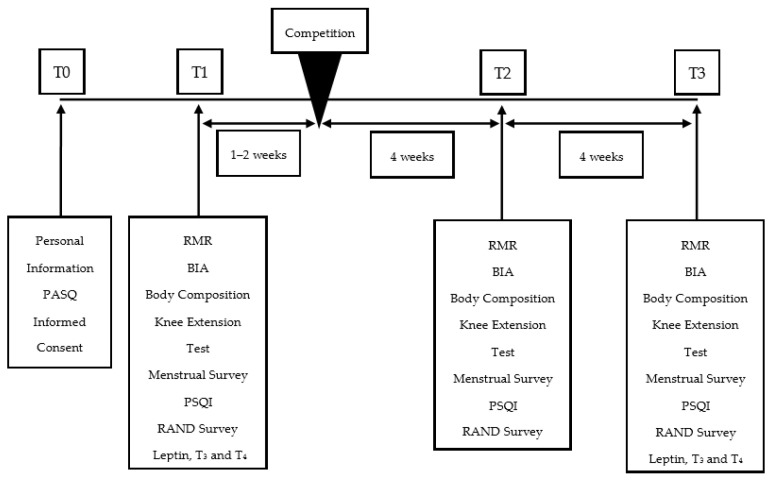
Overview of competitor assessment. RMR: Resting metabolic rate; BIA: Bioelectrical impedance analysis; PSQI: Pittsburgh sleep quality index.

**Figure 2 jfmk-05-00027-f002:**
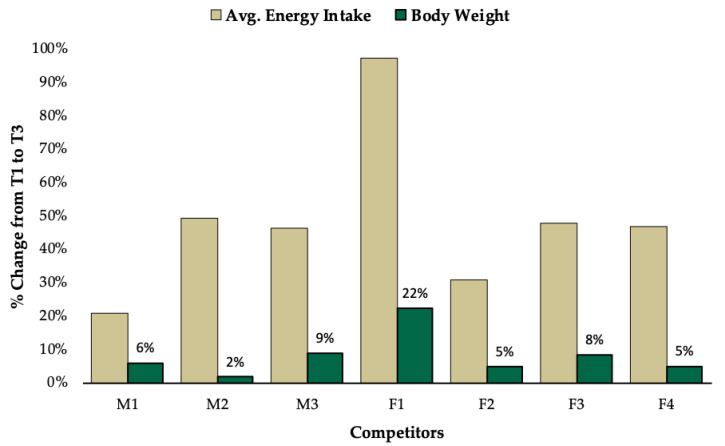
Percent change (%△) in average energy intake (kcal/day) and body weight (kg).

**Figure 3 jfmk-05-00027-f003:**
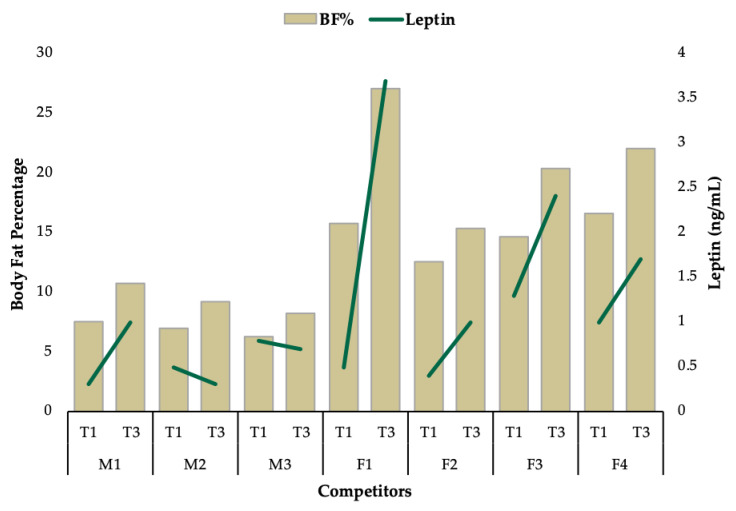
Changes in body fat percentage and leptin from T1 to T3.

**Figure 4 jfmk-05-00027-f004:**
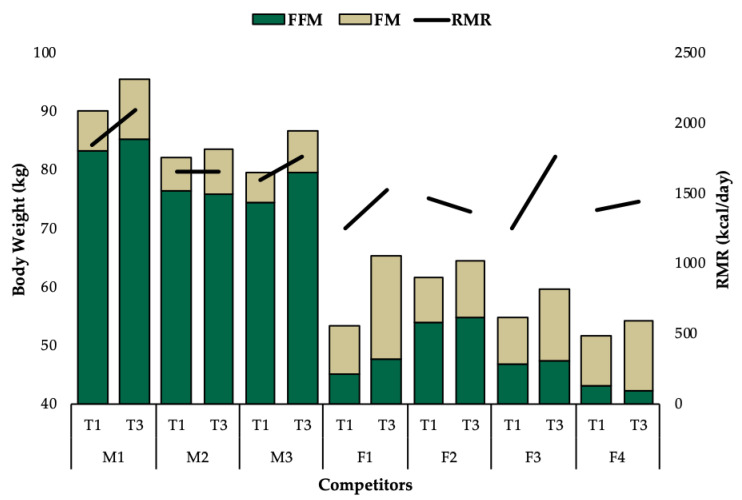
Individual responses for change in body weight and RMR from T1 to T3.

**Figure 5 jfmk-05-00027-f005:**
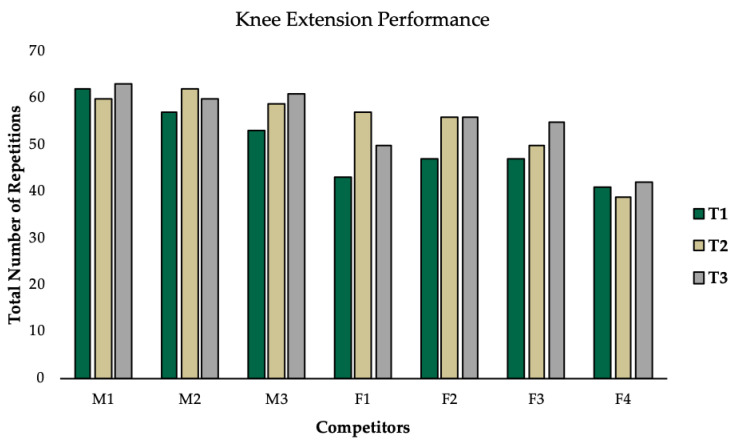
Individual responses of total number of repetitions completed during knee extension performance test.

**Table 1 jfmk-05-00027-t001:** Participant descriptive characteristics.

Competitor	Age	Height (cm)	Class	Contest Prep Duration (wk) ^1^	Precontest Prep Bodyweight (kg) ^1^
M1	42	191	Physique	20	102
M2	29	182	Physique	12	94
M3	32	173	Bodybuilding	39	94
F1	30	161	Figure	20	65
F2	29	170	Figure	17	70
F3	23	157	Bikini	12	57
F4	35	157	Bikini	25	57

^1^ Self-Reported.

**Table 2 jfmk-05-00027-t002:** Physiological variables.

		T1	T2	T3	T1–T3 Change
Male 1					
Body Composition	Body Weight (kg)	90.2	93.4	95.6	5.4
	Fat-Free Mass (kg)	83.4	84.8	85.4	2
	Fat Mass (kg)	6.8	8.6	10.3	3.5
	Total Body Water (kg)	60.8	62.7	61.6	0.8
	Dry Fat-Free Mass (kg)	22.6	22.1	23.8	1.2
Resting Metabolic Rate	REE (kcal/day)	1852	2013	2102	250
	RQ (a.u.)	0.79	0.91	1.03	0.24
	Predicted (kcal/day)	1978	2021	2053	75
Blood Hormones	Leptin (ng/mL)	0.3	-	1	0.7
	Free triiodothyronine (T_3_)(pg/mL)	2.7	-	2.7	0
	Free thyroxine (T_4_)(ng/dL)	1.1	-	1.1	0
Male 2					
Body Composition	Body Weight (kg)	82.3	82.3	83.8	1.5
	Fat-Free Mass (kg)	76.5	75.4	76.1	−0.4
	Fat Mass (kg)	5.7	6.9	7.7	2
	Total Body Water (kg)	55.1	54.9	55.3	0.2
	Dry Fat-Free Mass (kg)	21.4	20.5	20.8	−0.6
Resting Metabolic Rate	REE (kcal/day)	1660	1686	1667	7
	RQ (a.u.)	0.79	0.81	0.84	0.05
	Predicted (kcal/day)	1912	1912	1932	20
Blood Hormones	Leptin (ng/mL)	0.5	-	0.3	−0.2
	Free triiodothyronine (T_3_)(pg/mL)	1.8	-	1.8	0
	Free thyroxine (T_4_)(ng/dL)	0.9	-	0.9	0
Male 3					
Body Composition	Body Weight (kg)	79.6	84.3	86.8	7.2
	Fat-Free Mass (kg)	74.6	77.9	79.6	5
	Fat Mass (kg)	5.1	6.4	7.1	2
	Total Body Water (kg)	54.6	57.2	56.6	2
	Dry Fat-Free Mass (kg)	20	20.7	23	3
Resting Metabolic Rate	REE (kcal/day)	1601	1831	1765	164
	RQ (a.u.)	0.83	0.87	0.91	0.08
	Predicted (kcal/day)	1811	1875	1909	98
Blood Hormones	Leptin (ng/mL)	0.8	-	0.7	−0.1
	Free triiodothyronine (T_3_)(pg/mL)	1.9	-	2.4	0.5
	Free thyroxine (T_4_)(ng/dL)	0.8	-	1	0.2
Female 1					
Body Composition	Body Weight (kg)	53.6	64.7	65.6	12
	Fat-Free Mass (kg)	45.1	49.7	47.8	2.7
	Fat Mass (kg)	8.5	15	17.8	9.3
	Total Body Water (kg)	34.4	37.5	35.3	0.9
	Dry Fat-Free Mass (kg)	10.7	12.2	12.5	1.8
Resting Metabolic Rate	REE (kcal/day)	1264	1628	1536	272
	RQ (a.u.)	0.74	0.86	0.85	0.11
	Predicted (kcal/day)	1325	1431	1440	115
Blood Hormones	Leptin (ng/mL)	0.5	-	3.7	3.2
	Free triiodothyronine (T_3_)(pg/mL)	1.6	-	2.9	1.3
	Free thyroxine (T_4_)(ng/dL)	0.8	-	0.9	0.1
Female 2					
Body Composition	Body Weight (kg)	61.8	63.3	64.8	3
	Fat-Free Mass (kg)	54	53.8	54.9	0.9
	Fat Mass (kg)	7.7	9.5	9.9	2.2
	Total Body Water (kg)	42	42.4	43.9	1.9
	Dry Fat-Free Mass (kg)	12	11.4	11	−1
Resting Metabolic Rate	REE (kcal/day)	1471	1424	1381	−90
	RQ (a.u.)	0.76	0.83	0.8	0.04
	Predicted (kcal/day)	1426	1441	1455	29
Blood Hormones	Leptin (ng/mL)	0.4	-	1	0.6
	Free triiodothyronine (T_3_)(pg/mL)	1.5	-	2.4	0.9
	Free thyroxine (T_4_)(ng/dL)	0.8	-	1.1	0.3
Female 3					
Body Composition	Body Weight (kg)	55.1	57.5	59.7	4.6
	Fat-Free Mass (kg)	47	46.5	47.5	0.5
	Fat Mass (kg)	8	11	12.2	4.2
	Total Body Water (kg)	33.8	33.6	33.6	−0.2
	Dry Fat-Free Mass (kg)	13.2	12.9	13.9	0.7
Resting Metabolic Rate	REE (kcal/day)	1259	1486	1769	510
	RQ (a.u.)	0.84	0.85	0.91	0.07
	Predicted (kcal/day)	1366	1390	1411	45
Blood Hormones	Leptin (ng/mL)	1.3	-	2.4	1.1
	Free triiodothyronine (T_3_)(pg/mL)	2.8	-	3	0.2
	Free thyroxine (T_4_)(ng/dL)	1	-	1.1	0.1
Female 4					
Body Composition	Body Weight (kg)	51.8	54.1	54.5	2.7
	Fat-Free Mass (kg)	43.2	42.5	42.5	−0.7
	Fat Mass (kg)	8.6	11.6	12	3.4
	Total Body Water (kg)	31.9	32.4	31.7	−0.2
	Dry Fat-Free Mass (kg)	11.3	10.1	10.8	−0.5
Resting Metabolic Rate	REE (kcal/day)	1393	1482	1447	54
	RQ (a.u.)	0.82	0.86	0.8	−0.02
	Predicted (kcal/day)	1279	1301	1305	26
Blood Hormones	Leptin (ng/mL)	1	-	1	0
	Free triiodothyronine (T_3_)(pg/mL)	2.5	-	3	0.5
	Free thyroxine (T_4_)(ng/dL)	1	-	1.1	0.1

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
