# Peer review of "Physiological, Psychological and Performance-Related Changes Following Physique Competition: A Case-Series"

_jfmk, 2020, doi:10.3390/jfmk5020027_

Round 1

Reviewer 1 Report

The purpose of this case series was to evaluate the physiological, psychological and performance-related changes that occur during the post-competition period. The authors concluded that :

  • Physique competitors expecting to optimally recover from competition should include a level of energy intake that elicits a certain amount of body fat gain, while avoiding body fat overshooting.
  • Physique athletes are encouraged to incorporate longer off-seasons not only to allow for proper recovery of key variables, but to support improvements in lean body mass and metabolic rate for future competitive success.

The manuscript is well written and present interesting findings. However, some modifications are needed:

Introduction:

A short paragraph about overtraining is important for this section (see Polito et al. 2017; Science & Sports32(1), 1-13).

The effect of changes in parameter like sleep, diteray intakes, etc. on performance are important to introduce in this parts (see Boukhris et al. (2019; Sports7(5), 118.) and Trabelsi et al. (2019; British journal of sports medicine, bjsports-2018.).

I suggest adding an hypothesis.

Methods:

Do the authors perform a sample size calculation?

Results

Well written.

Discussion

I suggest adding a practical recommendations part at the end of this section.

Reviewer 2 Report

Thank you to the authors for an interesting paper, that was an engaging read and largely sound scientifically. I have provided line by line recommendations, and areas that I think the authors should consider expanding to inform readers to a greater extent, and further contextualise the work conducted.

My major concern is the quality of the figures presented, these changes are easily made and this will serve to 'tidy up' the manuscript, which is otherwise presented to a high quality and in accordance with the journal's guidelines. Please can the authors delete the border and horizontal lines for figures 2-5, ensure all text and axes lines are in black and consider changing font to Palatino Linotype in accordance with the rest of the text in the manuscript. For figure 1 please check the alignment within text boxes and consider increasing the line thickness of boxes and arrows to make these features stand out.

Please also check the spacing at the start of sentences - double spacing appears to be used throughout, but this is inconsistent with the conventions of the journal, and ensure that either the abbreviation M/F or full male/female is used consistently throughout the text to describe participants.

Line by line:

37: if only using one value to describe LEA i.e. <30kcal/kg/FFM, please revise criteria to criterion

37-40: this sentence may fit better earlier on, or consider reworking the paragraph. Currently it seems a little disjointed, as you go from physiological impairments to LEA and back to further discussion of the physiological impairments. 

45: 'these parameters' - in this context, having now discussed physiological and psychological parameters, are we reading about both phys and psyc symptoms or psychological symptoms only? Please be specific to make this more accessible for a reader with little background in this area

Line 46: 'highly variable between individuals' - have you a reference or series of references to support this? If you do, multiple references may work better, with a transition into the remainder of the paragraph which nicely details typical time courses of recovery for varying physiological parameters.

It may further strengthen the introduction to consider Relative Energy Deficiency in Sport, which effectively models and ties together a lot of the concepts you've introduced - I would encourage some further discussion of hormonal changes - as this will allow the reader to understand why you've measured what you have, when you have, within the methodology.

54: please can you add words to the effect of 'from physique competition' after 'the recovery phase' - whilst this is somewhat obvious, it may not be explicitly clear to readers for whom English is a second language or with little prior knowledge of physique competition who may be reading out of general interest

54 onwards - I agree that there is a need to investigate the post-competitive recovery phase, indeed your data and others suggest it's a very interesting phase with quite a wide range of responses seen. I would encourage the authors to reiterate the heterogeneity in responses here as a justification for assessing such responses in a case series as opposed to a single case study, as all that have incorporated the recovery phase to date appear to have been restricted to one individual. I feel you are underselling the work conducted somewhat.

57 - please state clearly the number of each sex you have examined, currently it can be interpreted as seven males and seven females.

105: please state the variability/reliability of the InBody system, as you have done for each other assessment - did you assess hydration levels by any other method prior to taking InBody measurements to ensure consistent preparation for the assessment? If this was not possible due to dietary requirements of each competitor then please state so.

121-123 - please can you confirm the test-retest reliability of your anthropometrist - my understanding is that this is typically expressed in mm, as opposed to %BF, as %BF is obviously derived from the equations you've referenced?

132 - 'a more favourable health state' - is this how the RAND questionnaire describes higher values?

157 - please double check author guidelines as to whether an operating address is required for the testing laboratory, given this is effectively a brand name

Section 2.8 - What tools/trackers were used to analyse these data? How were these data analysed?

169 -172: please rewrite this section to avoid repetition, as all delta operations are assessed/calculated in the same manner

Section 3.2 - you have quite rightly reported reliability/typical error for most assessments you have undertaken - it may be worth highlighting either in this section or in figures which participants have exceeded TE of tests, and thus more likely to have experienced a 'real change' over the course of their recovery e.g. Figure 4 would be a good reference for this, and may be further expanded upon in the Discussion.

Discussion - I have no revisions to report for the discussion, however I would like to see more of a 'so what' approach taken to key variables that you have assessed. I am particularly interested in the correlations between blood markers and body composition and a possible mechanistic explanation for this - are these changes consistent with other groups outside of physique competition?

Appendices - I think these may be better included in the text and expanded upon in the discussion - they highlight the variability of the individuals in their responses, and provide a much deeper understanding of the dietary approaches undertaken, which can then be further discussed in the discussion, in relation to the physiological results - the strength of a case series is that a lot of data can be discussed and easily viewed by the reader. More needs to be made of these data.

Thank you again for a clearly written case series on an interesting topic, whilst the changes above may appear somewhat onerous, I cannot stress enough that I enjoyed the manuscript and am impressed by the work that has been conducted - the manuscript and authors 'owe it to themselves' to reflect the quality and depth of the work conducted.

Round 2

Reviewer 1 Report

After this revision, I suggest that this version is suitable for publication.